# The molecular mechanism of snake short-chain α-neurotoxin binding to muscle-type nicotinic acetylcholine receptors

Mieke Nys [1,8] ✉, Eleftherios Zarkadas [2,3,8], Marijke Brams [1], Aujan Mehregan [1], Kumiko Kambara [4], Jeroen Kool [5], Nicholas R. Casewell [6], Daniel Bertrand [4], John E. Baenziger[7], Hugues Nury [2] & Chris Ulens [1] ✉

Bites by elapid snakes (e.g. cobras) can result in life-threatening paralysis caused by venom neurotoxins blocking neuromuscular nicotinic acetylcholine receptors. Here, we determine the cryo-EM structure of the muscle-type *Torpedo* receptor in complex with ScNtx, a recombinant short-chain α-neurotoxin. ScNtx is pinched between loop C on the principal subunit and a unique hairpin in loop F on the complementary subunit, thereby blocking access to the neurotransmitter binding site. ScNtx adopts a binding mode that is tilted toward the complementary subunit, forming a wider network of interactions than those seen in the long-chain α-Bungarotoxin complex. Certain mutations in ScNtx at the toxin-receptor interface eliminate inhibition of neuronal α7 nAChRs, but not of human muscle-type receptors. These observations explain why ScNtx binds more tightly to muscle-type receptors than neuronal receptors. Together, these data offer a framework for understanding subtype-specific actions of short-chain α-neurotoxins and inspire strategies for design of new snake antivenoms.

α-neurotoxins are peptide toxins that are abundantly present in the venom of elapid snakes and that competitively inhibit the nicotinic acetylcholine receptor (nAChR) to abolish normal neurotransmission thus leading to paralysis and subsequent respiratory failure. Such neurotoxic envenomings make a substantial contribution to the upwards of 138,000 snakebite deaths that occur each year, mostly in the tropics[1]. These α-neurotoxins belong to the superfamily of three-finger fold proteins (3FPs). 3FPs are small proteins composed of approximately 60–80 residues that adopt a common tertiary structure composed of three β-stranded loops (fingers) extending from a disulfide rich, hydrophobic core. Besides toxins, this family also includes

non-toxic proteins, like Lynx1, which is involved in the regulation of cholinergic transmission, and SLURP 1 and 2, which mediate cell proliferation[2–4]. However, the best characterized members of this superfamily are the snake three-finger fold toxins (3FTx). These toxins manifest a wide range of biological activities, including cytotoxicity, proteinase activity, and neurotoxicity[5,6]. This neurotoxicity mainly results from 3FTxs targeting the cholinergic system, including acetylcholinesterase, muscarinic receptors and nAChRs[7].

α-neurotoxins are high-affinity competitive inhibitors of the postsynaptic muscle-type nAChR, located at the neuromuscular junction[8]. According to their length and number of disulfide bonds, α-

[1]Laboratory of Structural Neurobiology, Department of Cellular and Molecular Medicine, Faculty of Medicine, KU Leuven, 3000 Leuven, Belgium. [2]University Grenoble Alpes, CNRS, CEA, IBS, F-38000 Grenoble, France. [3]University Grenoble Alpes, CNRS, CEA, EMBL, ISBG, F-38000 Grenoble, France. [4]HiQscreen, 1222 Vésenaz, Geneva, Switzerland. [5]AIMMS Division of BioMolecular Analysis, Vrije Universiteit Amsterdam, 1081 HV Amsterdam, Netherlands. [6]Centre for Snakebite Research & Interventions, Liverpool School of Tropical Medicine, L3 5QA Liverpool, UK. [7]Department of Biochemistry, Microbiology, and Immunology, University of Ottawa, Ottawa, ON K1H 8M5, Canada. [8]These authors contributed equally: Mieke Nys, Eleftherios Zarkadas. ✉e-mail: mieke.nys@kuleuven.be; chris.ulens@kuleuven.be

neurotoxins are classified into two main subfamilies: short-chain α-neurotoxins and long-chain α-neurotoxins. Short-chain α-neurotoxins are composed of 60–62 residues and are connected by four intramolecular disulfide bridges. Long-chain α-neurotoxins have 66–75 residues and have an additional disulfide bridge in finger II[7]. Although these α-neurotoxins share a high degree of sequence homology, they have distinct pharmacological profiles at nAChR subtypes. Both short- and long-chain α-neurotoxins inhibit muscle-type nAChRs with high affinity whereas only long-chain α-neurotoxins tightly bind to the neuronal α7 homopentameric nAChRs. Neuronal α4β2 and α3β2 receptors are mainly unaffected by α-neurotoxins[5].

One of the most intensively studied long-chain α-neurotoxins is α-Bungarotoxin (α-Bgtx), which was isolated in 1963 from the venom of the Taiwan banded krait *Bungarus multicinctus*[8] and led to the isolation, identification and purification of the *Torpedo* receptor, a muscle-type nAChR[9]. Furthermore, the structure-function relationship of long-chain α-neurotoxins and nAChRs has been explored intensively. This includes several crystal structures of α-Bgtx in complex with nAChR derivatives and homologs[10–12] as well as of α-cobratoxin, a long-chain α-neurotoxin from the Indo-Chinese spitting cobra *Naja siamensis* in complex with AChBP, a water-soluble homolog of the extracellular domain (ECD) of human nAChR[13]. More recently, high-resolution cryo-EM structures of both the heteropentameric *Torpedo* nAChR and the homopentameric α7 nAChR in complex with α-Bgtx were solved[14,15]. Based upon these structures the molecular determinants for long-chain α-neurotoxin binding to nAChRs were identified and include both N-linked glycans as well as loops (loop C and loop F) from the ligand-binding site, which is situated at the interface between receptor subunits in the ECD. Interestingly, the N-linked glycans involved in the accommodation of α-Bgtx at the neuronal α7 nAChR and muscle-type *Torpedo* nAChR are positioned at opposing sides of the ligand-binding pocket (the complementary and principal side, respectively). Furthermore, the highly conserved arginine residue at the tip of finger II is positioned behind loop C in both receptors, though in the case of the α7 nAChR finger II wedges deeper into the pocket possibly due to the steric hindrance caused by the extended Loop F in the *Torpedo* nAChR[14,15]. A wealth of structural and functional data is thus available for the interaction of long-chain α-neurotoxins with nicotinic acetylcholine receptors. In contrast, detailed insights into the molecular determinants of short-chain α-neurotoxin binding are lacking, despite these toxins being of great pathological relevance for causing systemic neurotoxicity during snakebite.

In this study, we take advantage of ScNtx, a short-chain α-neurotoxin with a consensus amino acid sequence based upon a multiple sequence alignment considering the eleven most toxic short-chain α-neurotoxins from elapid snakes in America, Africa, Asia and Oceania[16,17]. ScNtx can be produced in a recombinant manner in *E. coli*, thus it is a highly attractive tool for structural and functional studies. Additionally, ScNtx was designed to show better antigenic properties and thus to produce better experimental antivenoms in immunized animals[17]. Here, we perform an in-depth pharmacological characterization of this toxin and use cryogenic electron microscopy to determine the structure of ScNtx bound to the *Torpedo* receptor. Comparison of this structure with the long-chain bound structures provides a thorough understanding of the structural principles unique to the binding of short-chain α-neurotoxins. In combination with electrophysiological studies using mutants of ScNtx, this study offers a framework for understanding subtype-specific actions of short-chain α-neurotoxins.

## Results

### Cryo-EM structure of the *Torpedo* nAChR in complex with the consensus short-chain ScNtx

To investigate the affinity of ScNtx for the native, purified *Torpedo* muscle-type nAChR reconstituted in asolectin-MSP1E3D1 lipidic

nanodiscs, we employed microscale thermophoresis (MST) with a fluorescently labeled variant of ScNtx, ScNtx-NT647. Firstly, we performed a binding check to confirm the binding of ScNtx to the receptor (Fig. 1a). Subsequently, the change in normalized fluorescence as a function of *Torpedo* nAChR concentration was determined revealing an apparent affinity of 28 nM (Fig. 1b). This observation confirms the tight interaction of the short-chain ScNtx with muscle-type nAChRs. To further investigate the molecular basis for this interaction, we determined the structure of the *Torpedo* nAChR in complex with ScNtx, using single-particle cryo-electron microscopy (cryo-EM) in the presence of a megabody $Mb^{c7HopQ}$ that helped randomize orientations[18] (Supplementary Fig. 1a). The dataset allowed us to obtain a 3D reconstruction with an overall resolution of 3.15 Å and a local resolution better than 2.5 Å for the extracellular domain (ECD), including two bound toxins (Fig. 1c–f and Supplementary Fig. 1c–d, Table 1). The overall architecture is consistent with previously determined structures of the *Torpedo* nAChR and consists of five subunits, $\alpha_\gamma$-γ-$\alpha_\delta$-δ-β (Fig. 1c–f, Supplementary Fig. 2)[14,19]. When viewed from the top, these subunits are radially arranged in a counterclockwise manner around a central ion-conducting pore. Each subunit consists of an ECD, formed by a short N-terminal α-helix and ten β-strands, a transmembrane domain (TMD), composed of four transmembrane α-helices and an intracellular domain (ICD). The ICD is composed of the post-M3 helix (MX), the intracellular helix (MA) and a disordered linker between the third and fourth transmembrane helix. However, in our structure this ICD was not resolved due to lack of cryo-EM density, except for the MX helices of the $\alpha_\gamma$ and β subunits. The absence of the ICD, which might be due to its preferential interaction with the air-water interface, to disc reconstitution conditions or to natural flexibility, does not impact the ECD conformation and the binding of ScNtx.

When viewed from the side (Fig. 1e, f), the *Torpedo* nAChR-ScNtx complex resembles a T-shape with two toxin molecules extending nearly parallel to the membrane. Similar to the receptor in complex with α-(α-Bgtx)[14], two ScNtx molecules are bound at the neurotransmitter binding sites located at the $\alpha_\gamma$-γ interface and the $\alpha_\delta$-δ interface where they are stabilized by interactions with both the principal α (+) and complementary γ/δ (−) subunit. A superposition of the *Torpedo* receptor structures in the α-Bgtx onto the ScNtx-complex reveals a root mean square deviation (R.M.S.D) of 1.27 Å for C α atoms, indicating that the conformation of the *Torpedo* nAChR is virtually identical in both complexes. As α-Bgtx, ScNtx acts as a competitive antagonist by stabilizing the ion channel pore in a resting state[14].

### Comparison with the α-Bungarotoxin bound *Torpedo* nAChR structure reveals a different binding mode for long chain α-neurotoxins versus short-chain α-neurotoxins

ScNtx is composed of 60 residues adopting the typical three-finger fold of α-neurotoxins (Fig. 2a–e). This three-fingered hand consists of three β-stranded loops (fingers I–III) extending from a globular core containing four disulfide bridges (Fig. 2a–e and Supplementary Fig. 3). Both for ScNtx and α-Bgtx 24 % of their available surface area is buried by the receptor (1130 Å² for α-Bgtx, 1040 Å² for ScNtx) (Supplementary Table 2). Similar to α-Bgtx, loop C from the principal side (α subunit) and loop F from the complementary side (δ/γ subunit) pinch onto finger II of the toxin (Fig. 2b). This finger is positioned behind loop C which is in an extended conformation. Whereas the position of the conserved arginine residue at the tip of finger II (R31 in ScNtx, R36 in α-Bgtx) is similar in both toxins, a clear tilt of ScNtx is observed towards the complementary side (Fig. 2b, d, e and Supplementary Fig. 3). This results in the bottom part of loop F being tightly grasped between finger II and III of ScNtx. The tilt is most obvious when viewed from the base of the three-fingered hand, which is formed by loops connecting the different β-strands (Fig. 2d–e and Supplementary Fig. 3). When measured between ScNtx and α-Bgtx, after superposing both toxin receptor complexes, the displacement of these loops amounts to 7.9 Å

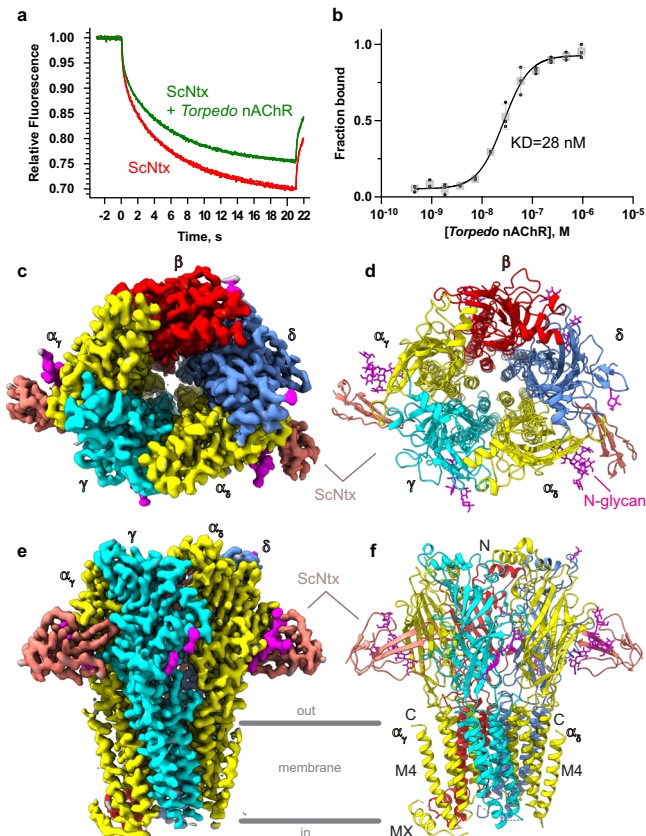

**Fig. 1 | Binding of ScNtx to the muscle-type *Torpedo* nAChR: biophysical and structural characterization. a** MST traces of fluorescently labeled unbound (red) and bound (green) ScNtx to purified *Torpedo* muscle-type nAChR reconstituted in asolectin-MSP1E3D1 lipidic nanodiscs. **b** MST concentration-response curve: change in normalized fluorescence as a function of *Torpedo* nAChR concentration. Data are presented as single data points in addition to the mean values ± standard deviations. **c, e** Top and side view of the cryo-EM map. **d, f** Top and side view of the 3D reconstruction in cartoon representation. N-glycans are shown as sticks. Source data are provided as a Source Data file.

for the β2–β3 loop, 6.7 Å for the β4-β5 loop and 10.4 Å for the β5-Cterm loop (Fig. 2e). In addition to this tilt, a 13 Å displacement of the tip of finger I is observed when measured between the two toxins (Fig. 2c and Supplementary Fig. 3). Consequently, finger I adopts a markedly different conformation in ScNtx and is in a position allowing equivalent interactions with the principal side as those formed by the extended C-terminus of α-Bgtx (Fig. 2c). Notwithstanding the displacement of finger I, it still stacks against the branched N-glycan attached to αN141 in the Cys-loop, like finger I of α-Bgtx[14]. Interestingly, these glycan residues adopt a different conformation in both structures allowing interactions with both ScNtx and α-Bgtx regardless of their different binding mode (Supplementary Fig. 4). This reinforces the vital role of N-linked glycan interactions in α-neurotoxin binding to nAChRs.

### The key role of loop F residues in accommodating short-chain α-neurotoxins

In addition to the different binding modes of ScNtx and α-Bgtx, the number and nature of the individual interactions within the neurotransmitter binding site differ substantially. An extensive analysis of these contacts was made by comparing the molecular interactions of both toxins with the receptor across the entire ligand binding site (Fig. 3, Table 1, Supplementary Movie 1). Here, we describe interactions with the principal (α_δ) and complementary (δ) subunit for finger I, II and III. The contact analysis at the α-γ interface is similar except for finger II of α-Bgtx which forms a higher number of interactions at the

complementary side than α-Bgtx at the α-δ interface. However, the overall conclusion remains the same for both interfaces (Table 1, Supplementary Table 3).

First, we zoom in on the interactions between finger I of ScNtx and α-Bgtx and the principal subunit (Fig. 3a, b). Finger I interacts with loop C, which was previously identified as a major molecular determinant of α-neurotoxin binding to nAChRs[20]. In the case of ScNtx, αW187, αY189 and αY190 as well as αT191 and αP194 that are situated at the tip of loop C are involved in this interaction. In contrast, the tip of finger I of α-Bgtx is in a different conformation and flipped away from the tip of loop C, thereby forming interactions that are limited to αW187 and αY189[14]. Remarkably, the extended C-terminus of α-Bgtx adopts a conformation similar to finger I of ScNtx, thereby forming equivalent interactions with residues at the tip of loop C, namely αY190, αT191, αP194 and an additional interaction with αC192. In summary, loop C residues on the principal side are involved in interactions that are roughly similar in the ScNtx versus α-Bgtx complex. However, the partnering residues at the level of the toxin differ due to the different conformation of finger I in α-Bgtx. This is compensated by the extended C-terminus of α-Bgtx, forming interactions comparable to finger I of ScNtx.

Next, we zoom in on the interactions between finger II and the principal (α_δ) subunit (Fig. 3c, d). One of the hallmark features present in all previously determined structures of nAChRs and AChBP homologs in complex with long-chain α-neurotoxins, is a unique cation-π interaction in which the highly conserved R36 in finger II is sandwiched between Y198 of loop C and F32 of α-Bgtx[14]. Interestingly, the side chain of R31 of ScNtx (equivalent to R36 in α-Bgtx) adopts a different rotamer conformation precluding the formation of a cation-π sandwich. Instead, R31 forms two H-bonds with αY190 in loop C. This αY190 adopts a key role in accommodating both ScNtx and α-Bgtx at the principal side. However, the nature of its interactions is markedly different. In the case of ScNtx it is involved in an extensive network of four H-bonds, two with R31, one with D29 and one with S8. By contrast, it is in an edge-to-face orientation relative to the cation-π sandwich formed in the α-Bgtx-bound structure and forms two H-bonds with D30 and H68 of α-Bgtx. In summary, the highly conserved arginine residue in finger II is involved in key interactions, namely a cation-π sandwich in the α-Bgtx complex and H-bonds in the ScNtx complex. This arginine residue plays a key role in anchoring the toxin deep into the neurotransmitter binding site and its position is virtually identical despite the different orientations of the two toxins.

Additionally, both toxins interact with the long, branched N-glycan derived from a conserved αN141 residue in the Cys-loop of the principal subunit through H-bonds (Fig. 3a–d). As described above, this N-glycan adopts a different conformation in both toxin-receptor complexes (Supplementary Fig. 4). However, the detailed conformation of each unique sugar molecule cannot be determined unambiguously. Depending on the position of the hydroxyl groups, H-bonds are formed with finger I or finger II of ScNtx.

On the complementary side (Fig. 3e–h), the interaction network of finger II-III is markedly different between the two toxins due to the tilt of ScNtx compared to α-Bgtx. Except for two residues of loop F, namely δL121, which interacts with F32 of α-Bgtx and δF184, which is involved in a H-bond with K45 of ScNtx, all interactions with finger II and III of the toxins involve the same six receptor residues for both toxins (Fig. 3e–h, Table 1), namely δT38 from loop G, δW57 from loop D and δD165, δD180, δP181 and δE182 of loop F. These six residues interact with seven different residues of α-Bgtx resulting in 13 Van der Waals (VdW) interactions. In contrast, nine different residues from ScNtx form 22 VdW interactions, one H-bond (between δE182 and K45) and three salt bridges (two between δD165 and R28 and one between δE182 and K25) with the same six receptor residues. This observation illustrates how the tilting motion of ScNtx toward the complementary side results in a much broader network of interactions. Thus, the

**Table 1 | Contacts between ScNtx/α-Bgtx and the α$_\delta$-δ interface of the *Torpedo* nAChR**

| | α-Bgtx | ScNtx | finger # |
|---|---|---|---|
| **Principal side (α)** | | | |
| W187 | **T6**, A7 | **Q7** | I |
| V188 | V39 | I34 | II |
| Y189 | T6, **T8**, **S9**, P10 | Q7, S8, Q10 | I |
| | V39, **V40** | | II |
| Y190 | | **S8** | I |
| | **H68** | | Cterm |
| | **D30**, R36, G37, K38 | **D29**, **R31**, T33, I34 | II |
| T191 | | **S8** | I |
| | H68, **K70** | | Cterm |
| | R36, **K38**, V40 | **T33** | II |
| C192 | R36 | | II |
| | **K70** | | Cterm |
| P194 | | S9, **Q10** | I |
| | H68, Q71 | | Cterm |
| Y198 | R36 | | II |
| N-glycan | **T6, A7** | Q7 | I |
| | | **E36** | II |
| **Complementary side (δ)** | | | |
| T38 | A31 | H30 | II |
| W57 | F32 | H30 | II |
| L121 | F32 | | II |
| D165 | S34 | **R28** | II |
| D180 | D30 | W27, D29 | II |
| P181 | C29 | R28 | II |
| | Y54 | K45, P46, G47, I48 | III |
| E182 | W28 | **K25**, W27 | II |
| | Y54 | **K45**, I48 | III |
| F184 | | **K45** | III |

H-bonds and salt bridges are shown in bold, Van der Waals interactions are in normal font.

complementary side, especially loop F, plays a key role in the tight interaction of ScNtx with the *Torpedo* nAChR.

Collectively, our analysis shows that ScNtx and α-Bgtx, both of which share a common three-finger fold, engage in substantially different interactions with the *Torpedo* nAChR, possibly explaining long- and short-chain specific interactions with muscle-type receptors.

### Functional characterization of structure-based mutants of ScNtx

We used automated two-electrode voltage clamp (HiClamp) to perform an in-depth pharmacological and electrophysiological characterization of ScNtx. We determined the concentration-inhibition relationship of ScNtx on different nAChR subtypes (Fig. 4a–c, Table 2). The inhibition by ScNtx of acetylcholine-induced currents is the most pronounced in case of the *Torpedo* muscle-type nAChR ($IC_{50} = 25$ nM). The human muscle-type and neuronal α7 nAChRs are inhibited with comparable $IC_{50}$ values (61 nM and 76 nM, respectively) (Fig. 4a, b). The current recovery after inhibition is vastly different. Only a slow and partial recovery is observed for the muscle-type nAChRs (Fig. 4a), whereas the α7 nAChR displays an instantaneous and complete recovery after inhibition (Fig. 4b), indicating a tighter binding mode for ScNtx on muscle-type receptors. No inhibition was observed on the α4β2 nAChR (Fig. 4c). This was unsurprising since ScNtx interacts with Y189 and P194 from the *Torpedo* nAChR. The equivalent residues in the α4 subunit (K189 and I196) confer α-Bgtx insensitivity on the neuronal α4β2 nAChR[21]. To further analyze the different binding modes in

muscle and α7 receptors, alanine-scanning mutagenesis was performed by mutating residues at the tip of the three toxin fingers. Mutation of finger I residues (S8A and S9A) prevented expression of ScNtx, indicating that these residues are crucial for protein folding or expression. In contrast, all other mutants of both finger II and finger III expressed readily and were purified in a biochemically stable state leading to a total of eight available ScNtx mutants.

To extrapolate our findings to human receptors we determined $IC_{50}$ values for all these mutants on both the human muscle-type and α7 nAChR. Striking differences were observed (Table 2). On the muscle-type receptor, these mutations have no effect, except for K25A (finger II), R31A (finger II) and K45A (finger III), which attenuate inhibition by ScNtx. In contrast, on α7 receptors all mutations either completely abolish or diminish inhibition by ScNtx. For example, the finger II mutation R28A has no effect on inhibition of the muscle receptor (Fig. 4d, h), but this mutation completely abolishes the inhibition of α7 receptors (Fig. 4e, I). Similar effects were observed for other finger II mutations, namely D29A, R31A (Fig. 4g, i) and G32A (Table 2). Additionally, on α7 receptors we observed an increase of the $IC_{50}$ values for K25A and H30A (finger II) and all finger III mutants (K45A, P46A and G47A). In contrast, most mutations do not cause a drastic change in the $IC_{50}$ value on muscle receptors, indicating a tighter interaction of ScNtx at these receptors. Only R31A (Fig. 4f, h), K25A and K45A exhibit an increased $IC_{50}$ value on muscle receptors. These results are consistent with the structural data. The highly conserved arginine residue at the tip of finger II, R31, penetrates deep into the binding site and is involved in four VdW interactions and two H-bonds with αY190 and as such a key determinant of short-chain α-neurotoxin binding to the principal subunit (Fig. 3c, Table 1). K25 and K45 are involved in 12 intermolecular contacts with the complementary side, including a salt bridge and two H-bonds (Fig. 3g, Table 1). It is thus not surprising that mutating K25, R31 and K45 causes a significant increase of the $IC_{50}$ values. By combining these three mutations in a triple mutant ScNtx (K25A + R31A + K45A), the inhibition of the muscle-type nAChR was completely abolished. We thus demonstrate that K25, R31 and K45, all highly conserved (Supplementary Fig. 5), are key determinants for short-chain α-neurotoxin binding to the muscle-type nAChR.

Interestingly, an increase of $IC_{50}$ values is not observed for R28 and D29, though both residues are involved in a number of interactions with the complementary and principal side, respectively. R28 forms two VdW interactions and two salt bridges with δD165 whereas the highly conserved D29 forms three VdW interactions and one H-bond with αY190 (Fig. 3e, c and Table 1). These latter interactions are possibly compensated by the tight interaction between αY190 with R31. All other mutations in ScNtx concern residues that are solely involved in VdW interactions and have no effect on muscle receptors. These mutations are likely compensated by the large network of interactions between ScNtx and the complementary face of the binding site.

In conclusion, we demonstrate two crucial differences in the properties of ScNtx on muscle-type versus the α7 nAChR. First, the recovery after inhibition by ScNtx is slow and only partial on muscle-type receptors, compared to an instantaneous and full recovery on α7 receptors (Fig. 4a, b). Second, the mutagenesis data demonstrate that mutations at the toxin-receptor interface have much more pronounced effects on α7 receptors than muscle-type receptors (Table 2). Combined, these results are consistent with a tighter interaction of ScNtx with muscle-type receptors, possibly explaining subtype-specific actions of short-chain α-neurotoxins on nAChRs.

This functional data was supplemented and substantiated by comparing the cryo-EM structure of the *Torpedo* nAChR to a model of the neuronal α7 nAChR, both in complex with ScNtx. This model was generated using AlphaFold2 (ptm score of 0.847 and pLDDT of 95.1) using the cryo-EM structure of the α7 nAChR in complex with α-Bgtx as

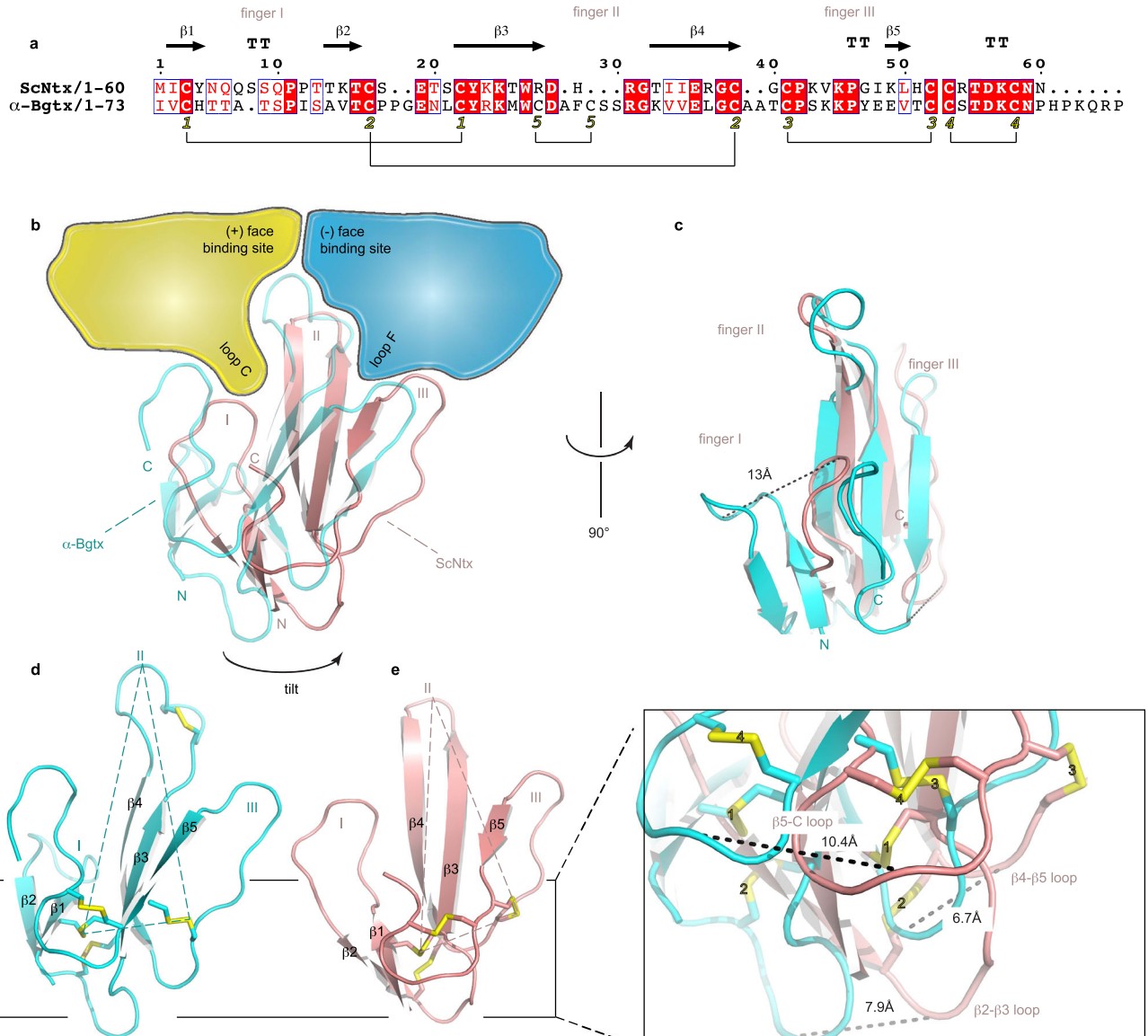

**Fig. 2 | Comparison of the ScNtx and α-Bgtx binding mode to the *Torpedo* nAChR. a** Sequence alignment of ScNtx and α-Bgtx generated in ESPRIPT[54]. Residue numbering and secondary structure information at the top is for ScNtx. TT indicates a β-turn. Yellow digits indicate disulfide bonds. Blue frames indicate regions of similarity. Red boxes indicate strict conservation, red characters indicate similarity. **b, c** Differences in binding mode of ScNtx (salmon) and α-Bgtx (cyan) at the α-δ subunit interface after superposition of the ScNtx-*Torpedo* nAChR complex onto the α-Bgtx-*Torpedo* nAChR complex (PDB ID: 6UWZ[14]). The principal subunit (α_δ) is shown in yellow; the complementary subunit (δ) is shown in blue. The orientations are chosen so that the tilt at the base of the three-fingered hand (**b**) and the conformational difference of finger I (**c**) are obvious. Roman numbers indicate the three fingers of ScNtx and α-Bgtx. **d, e** Cartoon representation of ScNtx and α-Bgtx. The orientation and color code are the same as in **b**. The disulfide bonds are shown in yellow. The tilt is indicated by dashed lines connecting the tip of finger II, which is in a fixed position, to disulfide bridges 1 and 3. The inset shows an enlarged view of the base of the three-fingered hand which is formed by loops connecting the different β-strands. These loops adopt markedly different positions in ScNtx (salmon) and α-Bgtx (cyan) as indicated by dashed lines.

custom template (PDB:7KOO) and contains ScNtx bound at the interface of two ECDs of the α7 nAChR[15,22] (Supplementary Data 1). The overall conformation of ScNtx is virtually identical (rmsd 0.567 Å) in the *Torpedo* and α7 nAChR bound structures (Supplementary Fig. 6). However, the binding mode differs substantially between both receptors. As is the case for α-Bgtx, the tip of finger II of ScNtx is wedged deeper into the binding pocket of the neuronal α7 nAChR (Supplementary Fig. 7). Additionally, due to the different length, sequence and conformation of loop F, only the tip of finger II approaches the complementary side close enough to allow H-bond formation (with R28). The contribution of finger III and the base of finger II thus seems negligible in the accommodation of ScNtx at the α7 nAChR. These data are in excellent accordance with the observed IC₅₀ values of the ScNtx mutants on both receptors and support a more localized interaction of ScNtx with the neuronal α7 nAChR compared to interactions that are more evenly spread across the different fingers in case of the muscle-type nAChR (Supplementary Fig. 6).

## Discussion

We determined the cryo-EM structure of the *Torpedo* nAChR in complex with the consensus short-chain α-neurotoxin, ScNtx (Fig. 1c–f). This study offers a framework for the understanding of high-affinity binding of short-chain α-neurotoxins to muscle-type nAChRs. Comparison with the long-chain α-neurotoxin α-Bgtx bound *Torpedo* nAChR demonstrates a drastically different binding mode for both toxins in the ligand-binding pocket (Fig. 2b–e). All three fingers adopt a different conformation which results in a profoundly different network of interactions (Fig. 3, Table 1, Supplementary Table 3). Both finger II

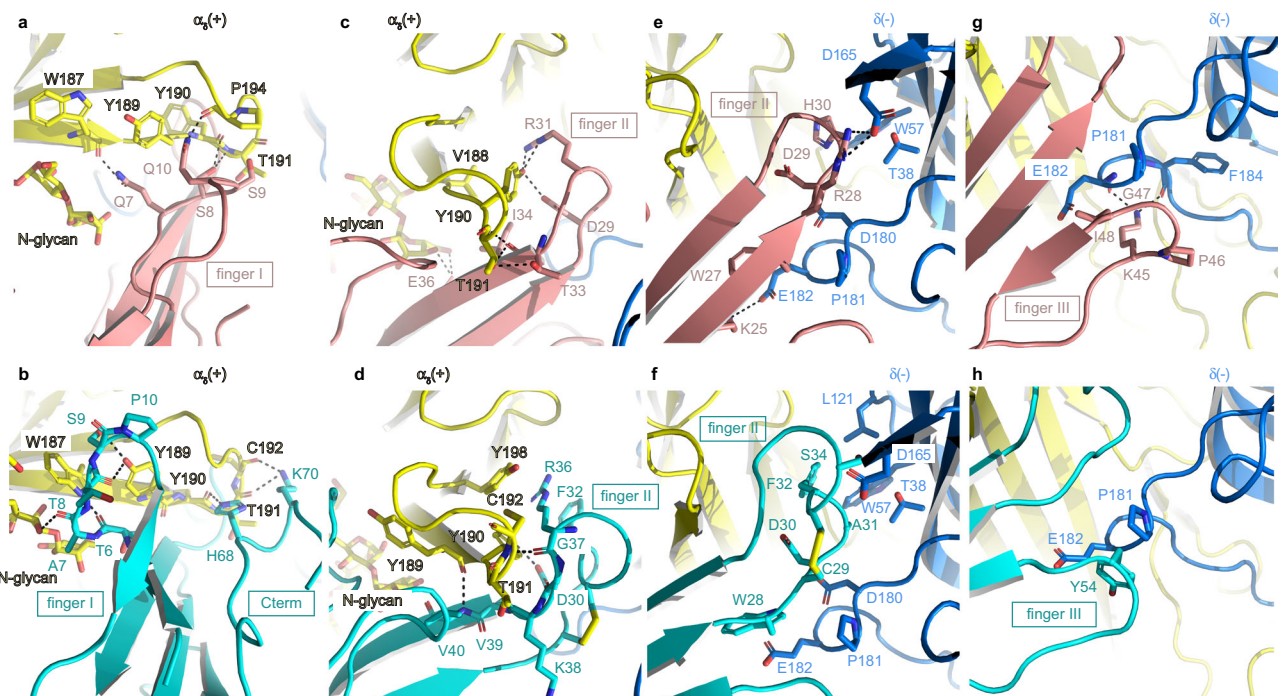

**Fig. 3 | Molecular contacts at the ScNtx and α-Bgtx-receptor interface.** Receptor and toxins are in cartoon representation. Interacting residues and glycans are shown as sticks, colored by subunit. Dashed lines indicate hydrogen bonds or salt bridges. Color code as in Fig. 2. The orientations are chosen so that the described interactions are clear (Supplementary Movie 1). The PDB ID for the α-Bgtx-*Torpedo* nAChR complex is 6UWZ[14]. **a, b** Interactions between the principal subunit (α$_δ$) and finger I of ScNtx and α-Bgtx. For α-Bgtx, additional interactions with the extended C-terminus are included. **c, d** Interactions between the principal subunit (α$_δ$) and finger II of ScNtx and α-Bgtx. **e, f** Interactions between the complementary subunit (δ) and finger II of ScNtx and α-Bgtx. **g, h** Interactions between the complementary subunit (δ) and finger III of ScNtx and α-Bgtx.

and III of ScNtx have shifted towards the complementary side resulting in loop F adopting a key role in the accommodation of short-chain α-neurotoxins as it accounts for 34% of the interaction surface versus 26% for α-Bgtx (Supplementary Table 2). This key role of loop F was further substantiated by an in-depth contact analysis and by site-directed mutagenesis studies revealing that the inhibition of the muscle-type nAChR by ScNtx was unaffected upon mutating interacting residues at the tip of finger II and III. In contrast, these mutations diminished or completely abolished the inhibition of the neuronal α7 nAChR by ScNtx (Figs. 3 and 4, Tables 1 and 2). This indicates that finger II and III residues from ScNtx are involved in a wider network of interactions with loop F residues from the muscle-type than from the α7 nAChR. Interestingly, both the sequence and length of loop F are poorly conserved among different nAChR subunits. Consequently, loop F was proposed to be a major determinant of nAChR subtype selectivity[23]. In contrast with the neuronal α7 nAChR, loop F from the δ- and γ-subunits adopts a more extended conformation, including an additional hairpin at the top of the toxin binding site[14,15] (Supplementary Fig. 4). The long-chain α-neurotoxin α-Bgtx is involved in a number of interactions with this variable loop F, but the number of interactions is much higher for the short-chain α-neurotoxin, ScNtx (10 versus 25). This possibly explains the different subtype selectivity of long- and short-chain α-neurotoxins. Since the network of interactions with loop F from muscle-type is extensive for short-chain α-neurotoxins and both the sequence and conformation of loop F is markedly different in the neuronal α7 nAChR, the binding of short-chain α-neurotoxins at the latter receptor will be less tight. In contrast, the more limited number of interactions between long-chain α-neurotoxins and loop F and the more pronounced role of the principal subunit in the binding of long-chain α-neurotoxins, allows a tight binding to both the muscle-type and the neuronal α7 nAChR. Indeed, the number of interactions between loop F residues from the α7 nAChR and α-Bgtx is limited (11 VdW interactions). However, the binding of α-Bgtx to the

complementary side of α7 nAChR is further stabilized by additional VdW interactions with loop D and E and by a H-bond with the N110-linked glycan[15]. We hypothesize that an equivalent H-bond with short-chain α-neurotoxins is impossible since this H-bond is formed with the extended tip of finger II of α-Bgtx which adopts a short helical motif. This motif is absent in the much more compact finger II of ScNtx and thus not within reach of the N110-linked glycan (Supplementary Fig. 4). This hypothesis is supported by the AlphaFold2 model of the α7 nAChR ECD in complex with ScNtx in which the tip of finger II is located lower in the binding pocket (Supplementary Data 1). However, there is an important limitation to our model. AlphaFold2 is currently unable to predict glycosylation. Therefore no glycan molecules are present in our model, nonetheless all cryo-EM structures indicate that they are involved in the accomodation of 3Ftxs at nAChRs[14,15]. Although we have validated the use of AlphaFold2 by modeling the *Torpedo* nAChR-ScNtx complex and comparing the resulting model to our cryo-EM structure, we emphasize that the AlphaFold2 model of the α7 nAChR ECD-ScNtx complex remains a hypothetical structure.

A detailed analysis of the contacts between the *Torpedo* nAChR and ScNtx combined with mutagenesis data, revealed the structural determinants of short-chain α-neurotoxin binding to muscle-type nAChRs. We demonstrated that both the principal and complementary side of the muscle-type nAChR are highly involved in this interaction, as well as all three fingers of ScNtx. This is in clear contrast with the limited contribution of loop F and finger III in the accommodation of long-chain α-neurotoxins at nAChRs[14,15]. Our data on the distinct role of finger III in the binding of short-chain neurotoxins to the muscle-type nAChR are in accordance with earlier EPR, fluorescence and NMR spectroscopy studies[24,25]. Both studies indicate K46 from neurotoxin II from *Naja oxiana* (equivalent to K45 in ScNtx) as a point of contact with the *Torpedo* nAChR. Our structure confirmed and supplemented this data by identifying the loop F residues (E182 and F184) involved in this interaction. The importance of loop F in the binding of short-chain

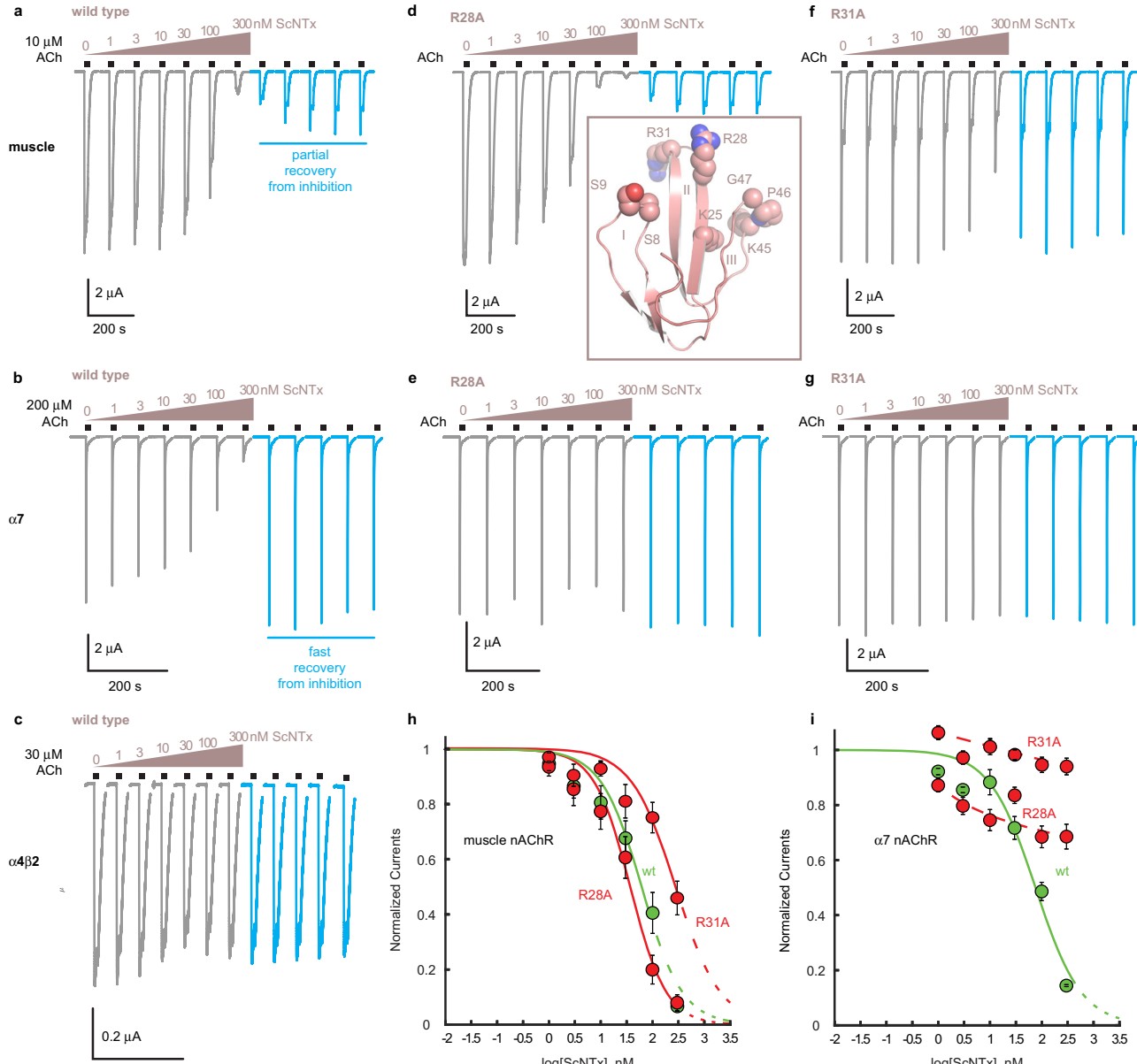

**Fig. 4 | Pharmacological characterization of ScNtx. a–g** Electrophysiological recordings on *Xenopus laevis* oocytes expressing different subtypes of nAChRs: the adult muscle-type nAChR, the neuronal α7 nAChR and the neuronal α4β2 nAChR. Horizontal bars indicate duration of acetylcholine application. Increasing concentrations of ScNtx or mutant ScNtx (R28A or R31A) were applied as indicated in salmon. Traces in salmon indicate co-application of acetylcholine (at a concentration near the EC50) and (mutant) ScNtx. Blue traces indicate application of acetylcholine alone. The inset depicts ScNtx in a cartoon representation. Residues subjected to mutagenesis are shown in spheres. Roman numbers indicate the three fingers. **h, i** Concentration-Inhibition curves: normalized currents in oocytes expressing the muscle-type nAChR or the neuronal α7 nAChR as a function of ScNtx concentration. Data are presented as mean values ± standard deviations. Green curves are for wild-type ScNtx, red curves for mutant ScNtx (R28A or R31A). Source data are provided as a Source Data file.

α-neurotoxins was recognized earlier[26,27]. NmmI, a short-chain α-neurotoxin from *Naja mossambica*, was shown to interact strongly with loop F from the mouse muscle-type nAChR. Moreover, pairwise mutagenesis identified the interaction between the highly conserved Lys from finger II (K25)[26,28] and loop F. A multipoint attachment of short-chain α-neurotoxins to nAChRs was thus suggested before[24,28]. Our structure confirms this hypothesis and provides a coherent structural context for understanding these results from previous studies.

We performed an in-depth pharmacological characterization of the consensus short-chain α-neurotoxin ScNtx. Similar to native short-chain neurotoxins, ScNtx is a high-affinity inhibitor of the human and *Torpedo* muscle-type nAChR. However, ScNtx also inhibits the neuronal α7 nAChR with an IC$_{50}$ value in the nanomolar range (Fig. 4a, b, Table 2). This is in contrast with certain native short-chain α-neurotoxins, which inhibit the α7 nAChR in the micromolar range[29,30]. In this respect, ScNtx is more similar to the native α-neurotoxin MlatA1 from the Mexican coral snake *Micrurus laticollaris* (82% identical to ScNtx), which also inhibits the α7 nAChR with affinity in the nanomolar range[31] (Supplementary Fig. 5). Moreover, both ScNtx and MlatA1 cause a slow and partial recovery after inhibition of the muscle-type receptor, whereas recovery is instantaneous and complete for the α7 receptor[31]. ScNtx thus maintains subtype-specific properties on different nAChR subtypes, displaying a tighter interaction with muscle-type receptors than α7 nAChRs. This pharmacological profile offered us the unique opportunity to investigate the molecular determinants

**Table 2 | IC50 values and Hill coefficients for wild-type ScNtx and mutants on the neuronal α7 nAChR and the muscle-type nAChR**

| | α7 nAChR | | | | | Human muscle-type nAChR | | | | |
|---|---|---|---|---|---|---|---|---|---|---|
| | IC50 (nM) | Hill coefficient | n | p value | | IC50 (nM) | Hill coefficient | n | p value | |
| ScNtx wt | 76.0 ± 13.2 | 0.98 ± 0.16 | 3 | | | 61.3 ± 19.8 | 1.11 ± 0.19 | 5 | | |
| **Finger I** | | | | | | | | | | |
| S8 | NE | | | | | | | | | |
| S9 | NE | | | | | | | | | |
| **Finger II** | | | | | | | | | | |
| K25A | 262.7 ± 20.2 | 2.65 ± 0.29 | 6 | 0.01 | * | 281.7 ± 23.6 | 1.18 ± 0.11 | 6 | 0.004 | ** |
| R28A | NI | | 8 | | | 37.3 ± 11.1 | 1.23 ± 0.35 | 5 | 0.20 | |
| D29A | NI | | 6 | | | 34.2 ± 5.8 | 1.66 ± 0.30 | 5 | 0.12 | |
| H30A | 120.4 ± 6.9 | 2.54 ± 0.18 | 5 | 0.05 | * | 12.5 ± 5.4 | 1.09 ± 0.13 | 4 | 0.02 | * |
| R31A | NI | | 6 | | | 278 ± 64 | 1.01 ± 0.12 | 4 | 0.01 | * |
| G32A | NI | | 7 | | | 79.2 ± 19.3 | 1.43 ± 0.13 | 6 | 0.43 | |
| **Finger III** | | | | | | | | | | |
| K45A | 124.7 ± 8.0 | 1.43 ± 0.05 | 6 | 0.03 | * | 247.2 ± 52.2 | 1.33 ± 0.15 | 6 | 0.005 | ** |
| P46A | 145.5 ± 10.4 | 1.85 ± 0.11 | 6 | 0.03 | * | 38.8 ± 3.3 | 1.31 ± 0.14 | 6 | 0.19 | |
| G47A | 391.0 ± 25.2 | 1.51 ± 0.49 | 5 | 0.03 | * | 89.5 ± 14.7 | 1.38 ± 0.10 | 6 | 0.22 | |
| **Triple mutant** | | | | | | | | | | |
| K25A + R31A + K45A | NI | | | | | NI | | | | |

Statistical significance for α7 nAChR was calculated between wild-type ScNtx and respective mutation using a two-tailed unpaired, non-parametric Mann–Whitney U-test. Statistical significance for muscle-type nAChR was calculated between wild-type ScNtx and respective mutation using a two-tailed unpaired, parametric t-test with Welch's correction for unequal standard deviations. */** Significantly different IC50 value from wild-type ScNtx. *$p < 0.05$, **$p < 0.005$. NE no expression of the mutant ScNtx in Origami 2 cells, NI no inhibition observed. Source data are provided as a Source Data file.

underlying the subtype specificity of short-chain α-neurotoxins through two-electrode voltage clamp on both nAChR subtypes using mutants of ScNtx. The results from these experiments show that certain mutations at the toxin-receptor interface eliminate inhibition of the neuronal α7 nAChR, but not of human muscle-type receptors (Fig. 4d–i, Table 2). This further substantiates subtype-specific actions of ScNtx on different nAChRs. In addition to ScNtx, other short-chain neurotoxins that recognize the neuronal α7 nAChR have been identified. For example, haditoxin isolated from the King cobra *Ophiophagus hannah* inhibits both the muscle-type and neuronal α7 nAChRs with the highest potency on the α7 nAChR[32]. A similar pharmacological profile is observed for fulditoxin isolated from the coral snake *Micrurus fulvius*[33]. However, both haditoxin and fulditoxin are homodimers composed of two short-chain α-neurotoxins whereas ScNtx adopts the prototypical monomeric conformation of short-chain α-neurotoxins.

The potential applications of ScNtx as a prototypical short-chain α-neurotoxin are further substantiated by the work of de la Rosa and colleagues[17]. They used ScNtx to obtain a horse-raised, experimental polyspecific antivenom. This antivenom successfully neutralizes the lethality of purified native and recombinant short-chain α-neurotoxins from different snake species in small animal models. This approach contrasts with currently available commercial antivenoms, which are all raised through the hyperimmunization of horses with one or more whole snake venoms. Although such antivenom therapies are clinically effective, they have several disadvantages, including high production costs, batch-to-batch variation and high incidences of adverse reactions[1,34]. Furthermore, antivenom has poor dose efficacy, with only 10–20% of resulting IgG antibodies being directed against the venom toxins used as immunogens, and an even lower proportion being directed against those toxins of greatest pathogenic importance[35,36]. For example, although α-neurotoxins are considered the most toxic components of elapid snake venoms, they are relatively small and weakly immunogenic resulting in a low number of IgG neutralizing these lethal toxins and suboptimal dose efficacy[37]. These deficiencies could be addressed by using recombinant consensus toxins during immunization, as ScNtx was designed with improved antigenic

properties, while immunization in the absence of other toxins that might dominate the immune response should help to improve the resulting dose efficacy[17]. Perhaps most importantly, current antivenom is highly snake species specific as venom toxin variation limits antibody cross-reactivity and neutralization[38]. Notwithstanding the potential advances in antivenom therapy offered by ScNtx, its cross-reactivity against several snake species remains relatively limited. A promising alternative pathway toward the development of anti-neurotoxin antivenom therapy with broad snake species cross-reactivity was recently proposed and consisted of a decoy-receptor approach[39]. Ligand fishing experiments demonstrated that the humanized α7/AChBP, a homolog of the ECD of the α7 nAChR, effectively captures certain venom toxins, mainly long-chain α-neurotoxins, from different snake species. Subsequent in vivo studies displayed an increased survival time of mice exposed to elapid venom upon treatment with α7/AChBP and a low dose of classical antivenom in comparison to animals receiving a low dose of antivenom alone. Thus, α7/AChBP holds therapeutic potential in the treatment of venomous snakebites. Other nAChR mimics, like the high affinity peptides (HAPs) which are mimotopes of loop C, are capable of preventing mice lethality caused by α-Bgtx toxicity[20]. However, so far both nAChR mimics, α7/AChBP and the HAPs, only capture long-chain α-neurotoxins, leaving the short-chain α neurotoxins, the most toxic components of many different elapid snake venoms[40], capable of causing paralysis and lethality. This is not surprising given the completely different binding mode of short-and long-chain α-neurotoxins at nAChRs revealed here in our study. Since α7/AChBP is a homolog of the human α7 nAChR, high-affinity and irreversible binding of short-chain α-neurotoxins is highly unlikely. The molecular determinants, identified in this study, provide a great source of inspiration for the rational engineering of α7/AChBP variants with increased affinity for short-chain α-neurotoxins, for example by mutating loop F.

In conclusion, we present the high resolution structure of a short-chain α-neurotoxin, ScNtx, bound to a native muscle-type nAChR. Comparison of this structure with the long-chain toxin complexes provides a thorough understanding of the structural principles unique

to the binding of short-chain α-neurotoxins. In combination with electrophysiological studies using mutants of ScNtx, we demonstrate that the binding mode of long- and short-chain α-neurotoxins is markedly different. Our data are consistent with a tighter interaction of short-chain α-neurotoxins with muscle-type receptors compared to neuronal α7 nAChRs. Together, these data offer a framework for understanding subtype-specific actions of short-chain α-neurotoxins and suggest strategies for the design of new snake antivenom with broad cross-species reactivity.

## Methods

### Purification of *Torpedo* nAChR

The nAChR was purified on a bromoacetylcholine bromide-derivatized Affi-Gel 102 column (Bio-Rad) as previously described[41], albeit with several modifications[19]. Briefly, crude membranes were solubilized in 1% of sodium cholate Tris Dialysis buffer (TDB: 100 mM NaCl, 10 mM Tris base, 1 mM EDTA, 0.02% $NaN_3$, pH 7.8) and a homemade cocktail of protease inhibitors. After ultracentrifugation (140'000 g for 30 min) to remove insoluble material, the supernatant was applied to the affinity column, washed with seven column volumes of 1% sodium cholate TDB supplemented with 1.05 mM soybean asolectin lipids (Sigma) and the bound protein eluted in 60 mL of the same buffer albeit at 250 mM NaCl and with 25 mM carbamylcholine (Sigma). The detergent solubilized nAChR was concentrated, treated with 15 mM dithiothreitol for 1 h to reduced disulfide-linked pentamers and further purified by size-exclusion chromatography on a Superose6 Increase column (GE healthcare). The purified pentameric nAChR was then concentrated to 2 mg/mL, incubated with MSP1E3D1 (Addgene plasmid # 20066[42]) at a 1:5 mol:mol ratio of nAChR to MSP3D1E3, and mixed with Bio-Beads (Bio-Rad) added to a final concentration of 400 mg/mL. The following day, the nAChR reconstituted asolectin-MSP1E3D1lipidic nanodiscs were purified by size-exclusion chromatography, concentrated to 0.65 mg/mL, aliquoted, snap frozen in liquid nitrogen and stored at −80 °C.

### Production and purification of ScNtx

We employed a synthetic gene (Genscript) encoding the full length ScNtx sequence[16] preceded by an N-terminal hexahistadine tag and a thrombin cleavage site. Codon usage was optimized for expression in *E. coli* and the gene was subcloned into the pQE-30 expression vector using *Bam*HI and *Pst*I restriction sites. This ScNtx/pQE-30 plasmid was transformed in *E. coli* K-12 derived Origami 2 cells (Novagen). Colonies containing ScNtx/pQE-30 were selected by their ability to grow on LB (Luria Bertani) plates containing 0.1 mg/ml carbenicillin (Fisher BioReagents). An initial expression test was performed by selecting individual colonies and resuspending them in 100 μl LB. 50 μl of this suspension was inoculated in 3 ml LB with 0.1 mg/ml carbenicillin, induced with 1 mM IPTG (isopropyl β-D-thiogalactoside) and incubated in a shaker-incubator overnight at 37 °C at 160 rpm. The remaining 50 μl was grown under identical conditions, but without IPTG and this was used as a negative control. Expression was verified by SDS-PAGE and Coomassie Brilliant Blue-R250 (Bio-Rad) staining. A positive colony was selected for overexpression in modified media (MM as published by de la Rosa et al., 2018[16]). For this purpose, 50 ml of LB with 0.1 mg/ml carbenicillin was inoculated with a positive colony and cells were grown overnight at 30 °C and 160 rpm. The next day 15 ml of this overnight culture was inoculated in 1 liter MM and grown at 37 °C, 160 rpm to log phase $OD_{600} = 0.6$–0.8 before inducing expression with 0.1 mM IPTG overnight at 16 °C. After collecting cells by centrifugation at 5000 x g for 15 min at 4 °C, they were lysed in extraction buffer BugBuster (Millipore) supplemented with 0.1 mg/ml lysozyme (Sigma), 90U Turbonuclease (Sigma) and cOmplete EDTA-free protease inhibitor cocktail (Roche) for 90 min with gentle agitation. Cell debris was removed by centrifugation at 20,000 x g for

30 min at 4 °C. The soluble ScNtx was purified from the supernatant by affinity chromatography. Therefore, the supernatant was supplemented with 25 mM imidazole pH 7.4 and loaded onto a 1 ml FF HisTrap column (Cytiva) pre-equilibrated in wash buffer A containing 1X PBS pH 7.4 (Sigma) plus 25 mM imidazole. After washing the HisTrap column with at least 50 column volumes of wash buffer A, the ScNtx protein was eluted by performing a linear gradient from buffer A to 1x PBS pH 7.4 plus 250 mM imidazole for 10 min at a flowrate of 1 ml/min. This was followed by a final elution step with 1x PBS pH 7.4 plus 500 mM imidazole. 1 ml fractions were collected and analyzed with SDS-PAGE (any kD™ Mini-PROTEAN TGX Precast gel, Bio-Rad) and Coomassie Brilliant Blue-R250 staining (Bio-Rad). Fractions containing ScNtx were pooled and concentrated to 0.5 ml using an Amicon centrifugal filter unit MWCO 3 kDa (Millipore). Next, to cleave off the hexahistidine tag we diluted ScNtx in 20 ml 1x PBS pH 7.4 and added 400 units of thrombin (Calbiochem) for overnight cleavage at 4 °C. The sample was then concentrated on an Amicon centrifugal filter unit MWCO 3 kDa (Millipore) to less than 1 ml. For a final polish purification step, the concentrate was loaded on a Superdex 75 10/300 (GE Healthcare) equilibrated with either 1x PBS pH 7.4 or 50 mM Tris-HCl pH 7.4, 150 mM NaCl. For quality control the peak fractions were run on an any kD™ Mini-PROTEAN TGX Precast gel (Bio-Rad) and visualized by Coomassie Brilliant Blue-R250 staining (Bio-Rad). The exact mass of the purified ScNtx was confirmed by high resolution mass spectrometry (7166.32 Da). The purified ScNtx was aliquoted and stored at −80 °C until further usage.

### Microscale thermophoresis

ScNtx was fluorescently labeled with NT647 using the Monolith NT™ Protein Labeling Kit RED-NHS (Nanotemper Technologies). The labeling was performed according to the manufacturer's instructions. Briefly, the fluorescent dye NT647 was dissolved in DMSO to a concentration of 435 μM and diluted to 60 μM in PBS pH 7.4. 100 μL of this dye solution was mixed with 100 μL of ScNtx in PBS pH 7.4 at a concentration of 20 μM (ratio dye: protein 3:1) and incubated in the dark at room temperature for 1 h. Unreacted free dye was removed by size-exclusion chromatography using column B from the kit. The degree of labeling was 0.77 moles dye per mole of ScNtx and was determined by measuring the protein absorbance at 280 nm (corrected for the fluorophore) and fluorescence absorbance at 647 nm. Labelled ScNtx (ScNtx-NT647) was then used for microscale thermophoresis experiments using the Monolith NT.Automated (Nanotemper technologies). Firstly, a binding check was performed using 5 nM ScNtx-NT647 and 2.5 μM purified *Torpedo* nAChR reconstituted in asolectin- MSP1E3D1 lipidic nanodiscs. The binding affinity (KD) was subsequently determined by analyzing the change in normalized fluorescence as a function of *Torpedo* nAChR concentration. Therefore, a serial dilution series was prepared with a concentration ranging between 0.5 nM and 1 μM *Torpedo* nAChR. All measurements were executed in triplicate and performed in PBS pH 7.4, 0.05% Tween at 20 °C, high MST power and 4 % excitation power. Monolith NT.Automated Premium Capillary Chips (Nanotemper technologies) were used in all experiments. Data processing and curve fitting was performed with the Nanotemper Analysis software (MO.Affinity Analysis v2.3).

### Oocyte preparation and injection

Preparation of the *Xenopus* oocytes was done using standard procedures and in agreement with animal care from Geneva Canton. Ovaries were harvested from deeply anesthetized females using cooling and MS-222 (5 g/L) and then sacrificed by section of the spinal cord and pitting. Ovaries were divided in two to three pieces and kept in sterile OR2 solution (88.0 mM NaCl, 2.5 mM KCl, 1.8 mM $CaCl_2.2H_2O$, 1.0 mM $MgCl_2$, 5.0 mM HEPES pH 7.85). A portion of the ovaries was

dissociated using mechanical and enzymatic procedure using Type-I collagenase (Sigma). Optimal expression of mRNA or DNA was obtained by injecting 15 nL of a solution containing 0.1 µg/µL using the automated Roboinject device (Multichannel System, Germany) as described in[43]. Oocytes were then incubated at 18 °C for two or more days before assessing the level of expression.

For α4β2 and α7 nAChRs, mRNA was prepared using the mMESSAGE mMACHINE™ T7 ULTRA Transcription Kit (Thermo-Fisher) following plasmid linearization with *Xma*I (ThermoFisher) and assessment of the quality on an agarose gel. For the expression of the neuromuscular junction (α1β1δε) receptors, cDNA injection of the plasmids in a 1:1:1:1 ratio was used with a final concentration of 0.04 µg/µL. For expression of the neuromuscular junction from *Torpedo*, the α1β1δγ subunits, injection of cDNA was conducted using a 1:1:1:1 ratio at a final concentration of 0.2 µg/µL.

### Electrophysiological recordings

After incubation, cells were investigated using the automated two-electrode voltage clamp system HiClamp (Multichannel System, Germany). Electrodes were pulled using the horizontal puller (SmartPull; UniPix, Switzerland) from borosilicate glass (1.2 mm O.D, 0.8 mm i.d.), filled with 3 M KCl and displayed a typical resistance of about 0.5 MOhms. Electrophysiological recordings were performed at 18 °C and cells were superfused with OR2 medium (see above). Data were acquired at 100 Hz; filtered at 20 Hz and analyzed using proprietary software running under Matlab (Mathworks Inc.). Cells were held at a potential of −80 mV and currents evoked by a concentration of ACh near the EC50 were recorded first in control and then in presence of a series of concentrations of the toxin applied in a growing order. Solutions were disposed in a 96 microtiter plate (NUNC, Thermofisher) and cells were incubated for 30 s before testing with the reference ACh test pulse (Fig. 4).

### Cryo-EM data collection and processing, model building

The *Torpedo* nAChR in asolectin-MSP1E3D1 nanodiscs was mixed with 10 µM of ScNtx and with the megabody $Mb^{c7HopQ}$ previously developed by Uchański et al.[18] (molar ratio receptor:megabody 1:3). This mega-body binds to the MSP scaffolding protein, not to the nAChR, and is present to help the nanodisc adopt multiple orientations. After an incubation time of 30 min on ice, 3.5 µL was deposited onto glow-discharged (30 mA, 50 s) Quantifoil Au/C R 1.2/1.3 grids. Blotting lasted for 6 s with force 0, at 8 °C and 100% humidity using a Mark IV Vitrobot (FEI, Thermo Fisher Scientific) before sample vitrification.

40-frame movies were recorded on a Gatan K2 Summit camera at the Glacios electron microscope of the IBS. The movies were imported to Cryosparc[44] which was used for all subsequent steps, except picking that was performed using crYOLO 1.7.6[45]. The box size was 256 pixels. After rounds of 2D classifications, classes showing pLGIC features were selected. Even with the presence of the megabody, top views remained more abundant and a rebalance of views was performed. The heterogeneity of the sample was reduced through an ab initio reconstruction and several rounds of heterogeneous refinement (Supplementary Fig. 1). The final set of particles was refined with Non-Uniform & local refinements[46].

The α-Bgtx-bound nAChR structure (6UWZ[14]) and a homology model of ScNtx based on the structure of erabutoxin a (5EBX[47]) prepared by SWISS-MODEL[48], were used as starting point and rigidly fitted in the map. Cycles of real-space refinement in Phenix[49] were performed, alternating with manual rebuilding in Coot[50]. Validation was performed with MolProbity[51] (Supplementary Table 1). The contact analysis was performed in ccp4[52]. Figures were prepared with PyMOL (Schrodinger) or ChimeraX[53].

A hypothetical model of the α7 nAChR in complex with ScNtx was predicted by AlphaFold2 using the cryo-EM structure of the α7 nAChR

in complex with α-Bgtx as custom template (7KOO) and the amino acid sequences of the ECD of the human α7 nAChR and ScNtx[15,16,22] (Supplementary Data 1).

### Statistics

Differences in IC$_{50}$ values for wild-type ScNtx and respective mutations of ScNtx were compared using a two-tailed unpaired, non-parametric Mann-Whitney U-test for the human α7 nAChR and using a two-tailed unpaired, parametric t-test with Welch's correction for unequal standard deviations for the human muscle-type nAChR.

### Reporting summary

Further information on research design is available in the Nature Research Reporting Summary linked to this article.

## Data availability

The coordinates and Cryo-EM map of the *Torpedo* nAChR in complex with ScNtx are deposited in the PDB under accession code 7Z14 and EMDB under accession code EMD-14440, respectively. The AlphaFold2 hypothetical model of the α7 nAChR in complex with ScNtx generated in this study was made available as Supplementary Data 1. PDB accession codes for structures referenced in this manuscript are: 6UWZ -*Torpedo* nAChR in complex with α-Bgtx and 7KOO - α7 nAChR in complex with α-Bgtx. Uniprot accession codes for protein sequences referenced in this manuscript are: human muscle nAChR subunit α P02708, β P11230, δ Q07001, ε Q04844 and γ P07510. Uniprot accession codes for short-chain α-neurotoxins: P01426, K9MCH1, P80548, P80958, P86095, P01434, P01418, P25675, P86420, P01424, P62388, Q45Z11, P01416, C1IC47, P60775, P60770. Source data are provided with this paper.

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

## Acknowledgements

This work was supported by grants KU Leuven C32/16/035 (CU) and the Wellcome Trust 221710/Z/20/Z (C.U., J.K. and N.R.C.). CU was supported by grants G0C1319N and G087921N from FWO-Vlaanderen. MN is a recipient of a FWO postdoctoral fellowship 12X2722N (M.N.). The microscale thermophoresis data were obtained at the KU Leuven Molecular Biophysics Platform of the SWITCH laboratory (Prof. Joost Schymkowitz and Prof. Frederic Rousseau) with support from Flanders Institute for Biotechnology (VIB, grant no. C0401); KU Leuven; and The Research Foundation - Flanders (FWO, equipment grant AKUL/15/34 - G0H1716N). This work used the EM facilities at the Grenoble Instruct-ERIC Center (ISBG; UAR 3518 CNRS CEA-UGA-EMBL) with support from the French Infrastructure for Integrated Structural Biology (FRISBI; ANR-10-INSB-05-02) and GRAL, a project of the University Grenoble Alpes graduate school (Ecoles Universitaires de Recherche) CBH-EUR-GS (ANR- 17-EURE-0003) within the Grenoble Partnership for Structural Biology. We thank Guy Schoehn and the IBS Electron Microscope facility, supported by the Auvergne Rhône-Alpes Region, the Fonds Feder, the Fondation pour la Recherche Médicale and GIS-IBiSA. This work was funded by the ERC Starting grant 637733 Pentabrain (HN). We thank Guillermo de la Rosa for his advice on the expression of an earlier version of the ScNtx expression plasmid.

## Author contributions

Conceptualization, M.N. and C.U.; Methodology, and Investigation, M.N., E.Z., M.B., A.M., K.K., J.K., N.R.C., D.B., J.E.B., H.N. and C.U.; Writing – Original Draft, M.N. and C.U.; Writing – Review & Editing, M.N., M.B., A.M., K.K., J.K., N.R.C., D.B., J.E.B., H.N. and C.U.; Funding Acquisition, M.N., H.N., J.K., N.R.C. and C.U.; Supervision, C.U.

## Competing interests

The authors declare no competing interests.
