## [Peer Review File · Nature Communications]

The molecular mechanism of snake short-chain α -neurotoxin binding to muscle-type nicotinic acetylcholine receptorsREVIEWER COMMENTS

Reviewer #1 (Remarks to the Author):

The major undoubtedly very valuable result of this work is the determination by cryo-EM of the three-dimensional structure of the complex between the muscle-type nicotinic acetylcholine receptor (nAChR) from the *Torpedo californica* and a snake venom alpha-neurotoxin of the short type (about 60 residues, 4 disulfides) with a good resolution, namely 2.5 Å for the portion of the extracellular domain where binding sites for the agonists and competitive antagonists (like snake venom neurotoxins) are situated. Apparently, the authors were very lucky that the “artificial” short-type neurotoxin was recently constructed and used as an excellent tool in producing very efficient antibodies as a protection against snake bites (references 16, 17 in the manuscript).

What is the true novelty of the presented manuscript if in the last two years cryo-EM structures were published both for the *Torpedo* and neuronal alpha7 nAChRs in complexes with another alpha-neurotoxin (references 14 and 15 in the manuscript)? The difference is essential –another alpha-neurotoxin is alpha-bungarotoxin belonging to the long-type neurotoxins (about 75 amino acid residues, 5 disulfides) which, contrary to the short toxins, efficiently inhibits not only the muscle-type receptors, but also the neuronal alpha7, as well as alpha9alpha10nAChRs. It was known that for the action on the alpha7 nAChR an important role is played by the additional (5th) disulfide in the central loop of the long-type three-finger alpha-neurotoxins. The importance of the work done and its novelty is due to the fact that all previous structures of the above-mentioned complexes, as well as of the models (ligand-binding domains of the alpha1 and alpha9 nAChR subunits or acetylcholine-binding protein) were solved only for the long-type alpha-neurotoxins alpha-bungarotoxin and alpha-cobratoxin.

The structure as such would be of interest because its comparison with the complexes of long neurotoxins already reveals essential differences in the binding modes of these two types of toxins: more strong interactions of the short toxin with the loop F at the complementary binding site. However, the authors have prepared a series of mutants of the used short toxin and with the electrophysiology methods identified its several residues important for binding to the *Torpedo* receptor. Those residues appear important for binding, although relatively weak, to the alpha7 nAChR, but their substitutions did not ruin strong binding to the *Torpedo* receptor. I would be glad to hear a more detailed explanation of this feature: important residues of the toxin were identified, as well as their partners in the receptor, but substitutions of these toxin residues are well tolerated.

Since snake-venom protein alpha-neurotoxins and peptide alpha-conotoxins are among the most efficient tools in research on nAChRs, quantification of the receptors at normal states and at diseases and also give hints for drug design, the obtained results are very important for the relevant fields. The previous literature in general is sufficiently well reflected in the manuscript. In fact, this work, by comparison of their own data and previous X-ray and cryo-EM structures, demonstrates the differences in the binding modes of the long-type and short-type neurotoxins. In particular, earlier for long neurotoxins not much role in binding appeared to play their loop III which was demonstrated to be essential for short toxins in the present work. However, I think that it would be appropriate to give

references to earlier work with “wet chemistry” where the participation of Lys 46 in loop III of short neurotoxin II *Naja naia oxiana* in binding to Torpedo nAChR was shown : Tsetlin V. Karlsson E et al. EPR and fluorescence study of interaction of *Naja naja oxiana* neurotoxin II and its derivatives with acetylcholine receptor protein from Torpedo marmorata. FEBS Lett. 1979;106(1):47. Moreover, similar conclusion follows from the data of solid-state NMR spectroscopy : Krabben L et al. Loop 3 of neurotoxin II is an additional interaction site with membrane bound nicotinic acetylcholine receptor... J. Mol. Biol. 390, 662-71 (2009)

What concerns the conclusions made on the basis of the solved structures and following mutagenesis-electrophysiology experiments, I have no critical remarks. However, I was surprised not finding a single word on the preparation and characteristics of the ScNtx-NT647 derivative – hope to see it in the revised manuscript.

I am not an electron microscopy specialist and cannot assess who reproducible can be the presented EM results, but I have no questions concerning heterologous expression of the toxin, its mutants and following testing of their activity.

Clearly, this work is a very important step in characterizing even small differences in the receptor binding modes of the different representatives of the same family of antagonists , giving new ways for characterizing different receptor subtypes and providing new tools for pharmacology and, hopefully, new leads to drugs.

Reviewer #2 (Remarks to the Author):

Summary-

The paper by Nys, et al. presents the cryo-EM structure of the Torpedo nicotinic acetylcholine receptor (AChR) in complex with the short-chain alpha-neurotoxin ScNtx; this toxin is apparently not found in nature but is based on a multiple sequence alignment of the most toxic short-chain toxins. The structure of the complex reveals differences between the complexes between the Torpedo or alpha7 AChRs and the long-chain neurotoxin alpha-bungarotoxin (Bgtx) published previously. In particular, whereas the long-chain neurotoxin interacts more extensively with the principal face of the orthosteric binding site, the short-chain ScNtx interacts more extensively with the complementary face of the site. The inter-residue contacts that stabilize the two toxins also differ, apparently due to differences in finger II. In detail, ScNtx, with four residues deleted from the tip of finger II, lacks a key Phe residue that in the complex with Bgtx participates in an arrangement of aromatic and cationic residues that includes two Tyr residues from the principal face and an Arg residue at the tip of finger II. In the absence of the key Phe, the Arg in finger II of ScNtx partners with a neighboring Asp residue to form an interaction with the aromatic hydroxyl group from one of the two key Tyr residues from the principal face, an apparently

weaker interaction that could enable a tilt to the complementary face. Conversely, the complex with ScNtx reveals multiple interactions between fingers II and III of the toxin and the complementary face, including hydrogen bonds and salt bridges; these are suggested to be a major source of stabilization of the complex. To test for sources of stabilization, candidate interacting residues in the toxin are mutated and the ability of the mutant toxins to block responses in muscle and alpha7 receptors is determined. Somewhat unexpectedly, mutations of some key interacting residues deduced from the structure do not reduce potency for the muscle AChR, and the most effective mutations in reducing potency do so by only around fourfold.

Critique-

The structure presented is the first of a short-chain alpha-neurotoxin with the Torpedo AChR, which is structurally and functionally analogous the human neuromuscular AChR. The conclusion that this short-chain alpha-neurotoxin interacts more extensively with the complementary compared to the principal face of the orthosteric site is reasonable. However, this conclusion is largely confirmatory. A short-chain neurotoxin was shown to interact strongly with the complementary face of the mouse muscle receptor (PMID: 10092644, PMID: 11112524), and one of the contacts identified in the present work (Glu182 from loop F with Lys25 of finger II) was identified by pairwise mutagenesis more than two decades ago (PMID: 10681526). This interaction was not tested, and the previous work was not referenced. In addition, the inability of ScNtx to block alpha4beta2 AChRs shown here can be rationalized by previous work (not referenced) showing that residues in alpha4 that juxtapose fingers I and II account for toxin insensitivity of these neuronal receptors (PMID: 31163179). This could have been cited in the results showing that the region between fingers I and II of the toxin interacts with multiple residues in loop C, two of which (Y189 & P194) are equivalent to those that confer insensitivity of alpha4beta2 to alpha-bungarotoxin. A major conclusion is that mutation of the Arg residue at the tip of finger II has a much greater reduction in potency for the alpha7 than the muscle AChR, suggesting a stronger interaction of ScNtx with the complementary compared to the principal face in the muscle AChR. However, none of the mutations of candidate residues reduce potency more than four-fold, and some key interacting residues apparent in the structure are not tested, such as Lys25 mentioned above. Multiple residues are likely responsible for ScNtx potency, though further work is needed to establish this. Finally, the author's suggestion that mimetic peptides based on loop F implicated in this work might neutralize the effects of short neurotoxins seems unlikely. An intact subunit interface, as in AChBP, perhaps including loop F of the muscle AChR would more likely adopt a native conformation and enable recognition of the short-chain neurotoxin.

Reviewer #3 (Remarks to the Author):

The manuscript by Nys and colleagues addresses the mechanisms of subtype specificity of binding of snake toxins to nicotinic acetylcholine receptors. For this the authors determined a cryo-EM structure of the muscle type receptor from *Torpedo* in complex with a consensus-sequence toxin ScNtx and demonstrated the interacting residues. Furthermore, they combined mutagenesis and electrophysiology to assess the effects of point mutations on the inhibition of muscle-type and neuronal-type nAChRs. This analysis demonstrated the critical importance of two residues R28 and R31 of ScNtx. The claim is that these data provide a thorough understanding of the structural principles unique to binding of short-chain alpha-neurotoxins and establishes the framework for understanding subtype-specific actions of short chain alpha neurotoxins.

The experiments are well performed and the data of high quality, however the actual framework for understanding the subtype-specificity also requires a model for in the interaction of the model toxin with a neuronal-type receptor. This is currently missing and the data does not fully support the claims, making the paper more narrowly focused than it could be. A cryo-EM structure or a good model of a neuronal receptor in complex with the toxin would definitely make the comparison more mechanistic and of a much higher general interest.

Other points:

1. The discussion about the potential peptides neutralizing toxins in the paragraph starting at line 376 does not seem convincing: the reviewer does not believe that it is possible to design peptides which could neutralize toxins with high enough efficiency. If the authors know the examples, they should refer to them in the manuscript.
2. The authors use megabodies linking to the nanodisc scaffold protein in order to allow the receptor to adopt multiple orientations. Strangely, I see mostly top views in Figure S1a. The reconstruction does not contain the ICD, which is present in some other structures of nAChR. Is this because the ICD is at the air-water interface? The authors should comment on the reason why the ICD is absent from the map and provide angular distribution for the reconstructions, for example in Figure S1.

Minor

3. Would displaying glycans in Figure 1 in different colors be beneficial for the readers?

REVIEWER COMMENTS

Reviewer #1 (Remarks to the Author):

The major undoubtedly very **valuable** result of this work is the determination by cryo-EM of the three-dimensional structure of the complex between the muscle-type nicotinic acetylcholine receptor (nAChR) from the *Torpedo californica* and a snake venom alpha-neurotoxin of the short type (about 60 residues, 4 disulfides) with a good resolution, namely 2.5 Å for the portion of the extracellular domain where binding sites for the agonists and competitive antagonists (like snake venom neurotoxins) are situated. Apparently, the authors were very lucky that the “artificial” short-type neurotoxin was recently constructed and used as an **excellent tool** in producing very efficient antibodies as a protection against snake bites (references 16, 17 in the manuscript). What is the true **novelty** of the presented manuscript if in the last two years cryo-EM structures were published both for the Torpedo and neuronal alpha7 nAChRs in complexes with another alpha-neurotoxin (references 14 and 15 in the manuscript)? The difference is essential –another alpha-neurotoxin is alpha-bungarotoxin belonging to the long-type neurotoxins (about 75 amino acid residues, 5 disulfides) which, contrary to the short toxins, efficiently inhibits not only the muscle-type receptors, but also the neuronal alpha7, as well as alpha9alpha10nAChRs. It was known that for the action on the alpha7 nAChR an important role is played by the additional (5th) disulfide in the central loop of the long-type three-finger alpha-neurotoxins. The **importance** of the work done and its **novelty** is due to the fact that all previous structures of the above-mentioned complexes, as well as of the models (ligand-binding domains of the alpha1 and alpha9 nAChR subunits or acetylcholine-binding protein) were solved only for the long-type alpha-neurotoxins alpha-bungarotoxin and alpha-cobratoxin.

We thank the referee for the positive comments and recognizing the value as well as the novelty of our work.

The structure as such would be of interest because its comparison with the complexes of long neurotoxins already reveals essential differences in the binding modes of these two types of toxins: more strong interactions of the short toxin with the loop F at the complementary binding site. However, the authors have prepared a series of mutants of the used short toxin and with the electrophysiology methods identified its several residues important for binding to the Torpedo receptor. Those residues appear important for binding, although relatively weak, to the alpha7 nAChR, but their substitutions did not ruin strong binding to the Torpedo receptor. I would be glad to hear **a more detailed explanation** of this feature: important residues of the toxin were identified, as well as their partners in the receptor, but substitutions of these toxin residues are well tolerated.

A similar point is also raised by referee #2, namely the fact that some mutants of ScNtx drastically reduce the affinity at the $\alpha 7$ nAChR, but certain mutants only very weakly (4-fold change at most) reduce the affinity at the muscle nAChR. This leads to a possible explanation that multiple residues could be responsible for ScNtx potency. To address this issue, we have performed additional experiments with the K25A mutant, as well as K25A+K45A+R31A triple mutant. In excellent agreement with the structural data the triple mutant completely eliminates inhibition of both the $\alpha 7$ and muscle nAChR by ScNtx. The K25A mutation significantly decreases inhibition of $\alpha 7$ and muscle nAChR by ScNtx. These results are now discussed on page 7, lines 268-292.

Since snake-venom protein alpha-neurotoxins and peptide alpha-conotoxins are among the most efficient tools in research on nAChRs, quantification of the receptors at normal states and at diseases and also give hints for drug design, the obtained results are very important for the relevant fields.

We thank the referee for recognizing the importance of our results in the context of toxins as research tools, receptor quantification and drug design.

The previous literature in general is sufficiently well reflected in the manuscript. In fact, this work, by comparison of their own data and previous X-ray and cryo-EM structures, demonstrates the differences in the binding modes of the long-type and short-type neurotoxins. In particular, earlier for long neurotoxins not much role in binding appeared to play their loop III which was demonstrated to be essential for short toxins in the present work. However, I think that it would be appropriate to give references to earlier work with “wet chemistry” where the participation of Lys 46 in loop III of short

neurotoxin II Naja naia oxiana in binding to Torpedo nAChR was shown : Tsetlin V. Karlsson E et al. EPR and fluorescence study of interaction of Naja naja oxiana neurotoxin II and its derivatives with acetylcholine receptor protein from Torpedo marmorata. FEBS Lett. 1979;106(1):47. Moreover, similar conclusion follows from the data of solid-state NMR spectroscopy :Krabben L et al. Loop 3 of neurotoxin II is an additional interaction site with membrane bound nicotinic acetylcholine receptor...J. Mol. Biol. 390, 662-71 (2009)

We thank the referee for directing our attention to these important references. These are now included and discussed in the manuscript on page 9-10, lines 384-396.

What concerns the conclusions made on the basis of the solved structures and following mutagenesis-electrophysiology experiments, I have no critical remarks. However, I was surprised not finding a single word on the preparation and characteristics of the ScNtx-NT647 derivative – hope to see it in the revised manuscript.

The referee makes a valid remark concerning the preparation of the ScNtx-NT647 derivative. To address this issue we have more extensively described the preparation of ScNtx-NT647 in the methods section on page 13, lines 545-552.

I am not an electron microscopy specialist and cannot assess who reproducible can be the presented EM results, but I have no questions concerning heterologous expression of the toxin, its mutants and following testing of their activity.

Clearly, this work is a **very important** step in characterizing even small differences in the receptor binding modes of the different representatives of the same family of antagonists , giving new ways for characterizing different receptor subtypes and providing new tools for pharmacology and, hopefully, new leads to drugs.

We thank the referee again for pointing out the importance of our work.

Reviewer #2 (Remarks to the Author):

Summary-

The paper by Nys, et al. presents the cryo-EM structure of the Torpedo nicotinic acetylcholine receptor (AChR) in complex with the short-chain alpha-neurotoxin ScNtx; this toxin is apparently not found in nature but is based on a multiple sequence alignment of the most toxic short-chain toxins. The structure of the complex reveals differences between the complexes between the Torpedo or alpha7 AChRs and the long-chain neurotoxin alpha-bungarotoxin (Bgtx) published previously. In particular, whereas the long-chain neurotoxin interacts more extensively with the principal face of the orthosteric binding site, the short-chain ScNtx interacts more extensively with the complementary face of the site. The inter-residue contacts that stabilize the two toxins also differ, apparently due to differences in finger II. In detail, ScNtx, with four residues deleted from the tip of finger II, lacks a key Phe residue that in the complex with Bgtx participates in an arrangement of aromatic and cationic residues that includes two Tyr residues from the principal face and an Arg residue at the tip of finger II. In the absence of the key Phe, the Arg in finger II of ScNtx partners with a neighboring Asp residue to form an interaction with the aromatic hydroxyl group from one of the two key Tyr residues from the principal face, an apparently weaker interaction that could enable a tilt to the complementary face. Conversely, the complex with ScNtx reveals multiple interactions between fingers II and III of the toxin and the complementary face, including hydrogen bonds and salt bridges; these are suggested to be a major source of stabilization of the complex. To test for sources of stabilization, candidate interacting residues in the toxin are mutated and the ability of the mutant toxins to block responses in muscle and alpha7 receptors is determined. Somewhat unexpectedly, mutations of some key interacting residues deduced from the structure do not reduce potency for the muscle AChR, and the most effective mutations in reducing potency do so by only around fourfold.

Critique-

The structure presented is the first of a short-chain alpha-neurotoxin with the Torpedo AChR, which is structurally and functionally analogous the human neuromuscular AChR. The conclusion that this short-chain alpha-neurotoxin interacts more extensively with the complementary compared to the principal

face of the orthosteric site is reasonable. However, this conclusion is largely confirmatory. A short-chain neurotoxin was shown to interact strongly with the complementary face of the mouse muscle receptor (PMID: 10092644, PMID: 11112524), and one of the contacts identified in the present work (Glu182 from loop F with Lys25 of finger II) was identified by pairwise mutagenesis more than two decades ago (PMID: 10681526). This interaction was not tested, and the previous work was not referenced. In addition, the inability of ScNtx to block alpha4beta2 AChRs shown here can be rationalized by previous work (not referenced) showing that residues in alpha4 that juxtapose fingers I and II account for toxin insensitivity of these neuronal receptors (PMID: 31163179). This could have been cited in the results showing that the region between fingers I and II of the toxin interacts with multiple residues in loop C, two of which (Y189 & P194) are equivalent to those that confer insensitivity of alpha4beta2 to alpha-bungarotoxin.

We thank the referee for pointing out these important references confirming our structure that we initially failed to discuss. These are now referenced and discussed on page 10, lines 390-396 and page 7, lines 258-261.

A major conclusion is that mutation of the Arg residue at the tip of finger II has a much greater reduction in potency for the alpha7 than the muscle AChR, suggesting a stronger interaction of ScNtx with the complementary compared to the principal face in the muscle AChR. However, none of the mutations of candidate residues reduce potency more than four-fold, and some key interacting residues apparent in the structure are not tested, such as Lys25 mentioned above. Multiple residues are likely responsible for ScNtx potency, though further work is needed to establish this.

To address the referee's concerns, we have conducted additional experiments and investigated the effect of the K25A mutant. Additionally, to investigate whether multiple residues are responsible for ScNtx potency we have investigated the effects of the K25A+R31A+K45A triple mutant. In excellent agreement with the structural data the triple mutant completely eliminates inhibition of both the $\alpha 7$ and muscle nAChR by ScNtx. The K25A mutation significantly decreases inhibition of $\alpha 7$ and muscle nAChR by ScNtx. These results are now discussed on page 7, lines 268-292.

Finally, the author's suggestion that mimetic peptides based on loop F implicated in this work might neutralize the effects of short neurotoxins seems unlikely. An intact subunit interface, as in AChBP, perhaps including loop F of the muscle AChR would more likely adopt a native conformation and enable recognition of the short-chain neurotoxin.

We agree with the referee's remark and we have modified the discussion about the AChBP-loop F variant. This is now mentioned in page 11, line 465.

Reviewer #3 (Remarks to the Author):

The manuscript by Nys and colleagues addresses the mechanisms of subtype specificity of binding of snake toxins to nicotinic acetylcholine receptors. For this the authors determined a cryo-EM structure of the muscle type receptor from Torpedo in complex with a consensus-sequence toxin ScNtx and demonstrated the interacting residues. Furthermore, they combined mutagenesis and electrophysiology to assess the effects of point mutations on the inhibition of muscle-type and neuronal-type nAChRs. This analysis demonstrated the critical importance of two residues R28 and R31 of ScNtx. The claim is that these data provide a thorough understanding of the structural principles unique to binding of short-chain alpha-neurotoxins and establishes the framework for understanding subtype-specific actions of short chain alpha neurotoxins.

The experiments are well performed and the data of high quality, however the actual framework for understanding the subtype-specificity also requires a model for in the interaction of the model toxin with a neuronal-type receptor. This is currently missing and the data does not fully support the claims, making the paper more narrowly focused than it could be. A cryo-EM structure or a good model of a neuronal receptor in complex with the toxin would definitely make the comparison more mechanistic and of a much higher general interest.

We thank the referee for recognizing the high quality of our data. We fully agree that a model for the interaction of the model toxin with a neuronal-type receptor would expand the scope of the work and provide a more mechanistic interpretation of the data. Since we could not obtain a stable complex of

$\alpha 7$ /AChBP with ScNtx and were thus unable to elucidate a three-dimensional structure of this complex, we took advantage of the powerful tool of AlphaFold2 (Jumper *et al.*, Nature, 2021) to generate a model of the neuronal $\alpha 7$ nAChR in complex with ScNtx (the coordinates are uploaded as supplementary information). AlphaFold is based on machine learning and is used to predict protein structures with high accuracy. To validate our approach, we first investigated whether AlphaFold2 could accurately predict the complex of the *Torpedo* nAChR with ScNtx. Because AlphaFold2 run through Colab notebooks could not handle predictions of the whole receptor pentamer, we limited the prediction to the sequences of the extracellular domains (ECD's) of the α/δ subunits and the ScNtx sequence in combination with the published *Torpedo* nAChR+ α -Bgtx resting state structure (pdb code 6UWZ) as custom input for the search. AlphaFold2 generated a prediction of the α/δ interface + ScNtx with astonishing accuracy when compared to our experimental cryo-EM structure (rmsd 0.652 Å), thereby validating our approach using AlphaFold2. Next, we generated a model for the neuronal $\alpha 7$ nAChR based on the $\alpha 7/\alpha 7$ ECD interface sequences, the ScNtx sequence and the published $\alpha 7$ nAChR+ α -Bgtx resting state structure (pdb code 7KOO) as custom input for the search. This produced a model for the $\alpha 7/\alpha 7$ ECD interface with one ScNtx molecule bound. This model provided important insight into the molecular determinants of ScNtx binding to the $\alpha 7$ nAChR ECD. Specifically, the tip of finger II of ScNtx is wedged deeper into the binding pocket of the neuronal $\alpha 7$ nAChR. Additionally, due to the different length, sequence and conformation of loop F, only the tip of finger II approaches the complementary side close enough to allow H-bond formation (with R28). The contribution of finger III and the base of finger II thus seems negligible in the accommodation of ScNtx at the $\alpha 7$ nAChR. These data are in excellent accordance with the observed IC50 values of the ScNtx mutants on both receptors and support a more localized interaction of ScNtx with the neuronal $\alpha 7$ nAChR compared to interactions that are more evenly spread across the different fingers in case of the muscle-type nAChR. Although we have validated the use of AlphaFold2, we emphasize that the model of the $\alpha 7$ nAChR ECD-ScNtx complex remains a hypothetical structure and has important limitations, including the absence of glycan molecules.

Other points:

1. The discussion about the potential peptides neutralizing toxins in the paragraph starting at line 376 does not seem convincing: the reviewer does not believe that it is possible to design peptides which could neutralize toxins with high enough efficiency. If the authors know the examples, they should refer to them in the manuscript.

We did refer to an important study by Harel *et al.* (Neuron, 2001) in which a high affinity peptide (HAP) was developed with picomolar binding affinity against α -Bgtx. However, we understand the referee's criticism and therefore we have omitted statements on loop F peptides.

2. The authors use megabodies linking to the nanodisc scaffold protein in order to allow the receptor to adopt multiple orientations. Strangely, I see mostly top views in Figure S1a. The reconstruction does not contain the ICD, which is present in some other structures of nAChR. Is this because the ICD is at the air-water interface? The authors should comment on the reason why the ICD is absent from the map and provide angular distribution for the reconstructions, for example in Figure S1.

The reviewer is right that mostly top views pop out of the Fig. S1a micrograph. There are two reasons for that. First, top views are much more recognizable, and thus even with a perfectly isotropic distribution of views the eye tends to focus on them. Second and more importantly, the distribution is far from being isotropic. This has been an observation, and sometimes an obstacle for structure resolution, in every nAChR nanodisc dataset we have collected so far. The addition of the megabody does help to rebalance the distribution, to some extent. In the dataset reported here, we further rebalance the distribution of views after 2D classification, as already noted in Fig S1b, throwing away ~130k top or close-to-top views. The angular distribution of the final set of particles, which remains anisotropic, is now shown.

The cause of the preferential orientation is uncertain. The reviewer suggests an eventual link to the ICD "liking" the air-water interface. It is possible, yet difficult (or at least it would be very time consuming) to ascertain. In Ryan Hibb's lab structures of the same receptor, the ICD is more resolved but there are multiple differences in the sample preparation and data treatment that could contribute to this difference: a different scaffold protein for the disc, a largely different protein concentration, the addition of the tensioactive Fos-Choline 8 to promote side views, and specific selection of ICD-containing particles

during data treatment. Adsorption at the air-water interface, in our case, could be prevented by the Fos-Choline, in their case. Or it could be that the different discs confer different levels of flexibility to the ICD, or that some Fos-Choline molecules enter the bilayer and modify its properties. How dynamic the ICD is when the receptor is in a cell remains to be explored. Because we unfortunately lack the microscope time to thoroughly test different environments/conditions, we have to be pragmatic. Thus, our reasoning was that the global nAChR conformation is that of the inhibited state, and we can safely interpret the binding of ScNtx.

The manuscript now states:

- Even with the presence of the megabody, top views remained more abundant and a rebalance of views was performed during data treatment. (page 14, lines 610-611)
- The absence of the ICD, which might be due to its preferential interaction with the air-water interface, to disc reconstitution conditions or to natural flexibility, does not impact the ECD conformation and the binding of ScNtx. (page 4, lines 134-136)

Minor

3. Would displaying glycans in Figure 1 in different colors be beneficial for the readers?
The glycans are now shown in a different color, as suggested by the referee.

REVIEWERS' COMMENTS

Reviewer #1 (Remarks to the Author):

The authors positively reacted to comments and critical remarks of all reviewers, including mine. It concerned the addition of the references reflecting the differences between the interactions of the short-type and long-type alpha-neurotoxins with the nicotinic acetylcholine receptors and also making corrections to some figures and introducing the new ones.

Reviewer #2 (Remarks to the Author):

The revised paper addresses concerns noted in the review of the initial submission as follows.

1. Previous work demonstrating interaction of short-chain neurotoxins with loop F is now recognized. The current work reveals this interaction at the level of inter-residue interactions.
2. Mutations of three residues in the short-chain neurotoxin that interact with loop F abolish toxin activity.
3. The suggestion that free peptides derived from loop F might be good anti-venoms is removed.

Two further corrections should be made as follows.

1. On line 65 it is stated that neurotoxin binding is irreversible. This statement is not correct. The binding is slowly reversible, and depends on species; in humans α -bungarotoxin dissociates from the adult muscle receptor with a half life of 5-6 hours, whereas in mouse or rat the half life is more than twice as long.
2. On line 66 it is said that long chain neurotoxins have high affinity for $\alpha 7$ and $\alpha 9$ homopentameric receptors. This is correct for $\alpha 7$ homomers, but for $\alpha 9$ is not certain. The reason is that there are only a few studies on $\alpha 9$ homomers because they don't express in many heterologous systems, and I don't know of any study showing high toxin affinity for $\alpha 9$. Better to stick with the statement about $\alpha 7$.

Reviewer #1 (Remarks to the Author):

The authors positively reacted to comments and critical remarks of all reviewers, including mine. It concerned the addition of the references reflecting the differences between the interactions of the short-type and long-type alpha-neurotoxins with the nicotinic acetylcholine receptors and also making corrections to some figures and introducing the new ones.

We would like to express our sincere gratitude for the reviewer's supportive attitude.

Reviewer #2 (Remarks to the Author):

The revised paper addresses concerns noted in the review of the initial submission as follows.

1. Previous work demonstrating interaction of short-chain neurotoxins with loop F is now recognized. The current work reveals this interaction at the level of inter-residue interactions.
2. Mutations of three residues in the short-chain neurotoxin that interact with loop F abolish toxin activity.
3. The suggestion that free peptides derived from loop F might be good anti-venoms is removed.

We thank the referee for taking the time to review the revised manuscript in detail.

Two further corrections should be made as follows.

1. On line 65 it is stated that neurotoxin binding is irreversible. This statement is not correct. The binding is slowly reversible, and depends on species; in humans alpha-bungarotoxin dissociates from the adult muscle receptor with a half life of 5-6 hours, whereas in mouse or rat the half life is more than twice as long.
2. On line 66 it is said that long chain neurotoxins have high affinity for $\alpha 7$ and $\alpha 9$ homopentameric receptors. This is correct for $\alpha 7$ homomers, but for $\alpha 9$ is not certain. The reason is that there are only a few studies on $\alpha 9$ homomers because they don't express in many heterologous systems, and I don't know of any study showing high toxin affinity for $\alpha 9$. Better to stick with the statement about $\alpha 7$.

We would like to thank the referee for these valid remarks. We adjusted the manuscript accordingly. The manuscript now states:

“Both short- and long-chain α -neurotoxins inhibit muscle-type nAChRs with high affinity whereas only long-chain α -neurotoxins tightly bind to the neuronal $\alpha 7$ homopentameric nAChRs.”